# ADAPTIVE LEARNING RATES FOR MULTI-AGENT RE-INFORCEMENT LEARNING

## ABSTRACT

In multi-agent reinforcement learning (MARL), the learning rates of actors and critic are mostly hand-tuned and fixed. This not only requires heavy tuning but more importantly limits the learning. With adaptive learning rates according to gradient patterns, some optimizers have been proposed for general optimizations, which however do not take into consideration the characteristics of MARL. In this paper, we propose AdaMa to bring adaptive learning rates to cooperative MARL. AdaMa evaluates the contribution of actors' updates to the improvement of Q-value and adaptively updates the learning rates of actors to the direction of maximally improving the Q-value. AdaMa could also dynamically balance the learning rates between the critic and actors according to their varying effects on the learning. Moreover, AdaMa can incorporate the second-order approximation to capture the contribution of pairwise actors' updates and thus more accurately updates the learning rates of actors. Empirically, we show that AdaMa could accelerate the learning and improve the performance in a variety of multi-agent scenarios, and the visualizations of learning rates during training clearly explain how and why AdaMa works.

## 1 INTRODUCTION

Recently, multi-agent reinforcement learning (MARL) has been applied to decentralized cooperative systems, *e.g.*, autonomous driving (Shalev-Shwartz et al., 2016), smart grid control (Yang et al., 2018), and traffic signal control (Wei et al., 2019). Many MARL methods (Lowe et al., 2017; Foerster et al., 2018; Rashid et al., 2018; Iqbal & Sha, 2019; Son et al., 2019) have been proposed for multi-agent cooperation, which follow the paradigm of centralized training and decentralized execution. In many of these methods, a centralized critic learns the joint Q-function using the information of all agents, and the decentralized actors are updated towards maximizing the Q-value based on local observation.

However, in these methods, the actors are usually assigned the same learning rates, which is not optimal for maximizing the Q-value. This is because some agents might be more critical than others to improving the Q-value and thus should have higher learning rates. On the other hand, the learning rates of actors and critic are often hand-tuned and fixed, and hence require heavy tuning. More importantly, over the course of training, the effect of actors and critic on the learning varies, so the fixed learning rates will not always be the best at every learning stage. The artificial schedules, *e.g.*, time-based decay and step decay, are pre-defined and require expert knowledge about model and problem. Some optimizers, *e.g.*, AdaGrad (Duchi et al., 2011), could adjust the learning rate adaptively, but they are proposed for general optimization problems, not specialized for MARL.

In this paper, we propose AdaMa for adaptive learning rates in cooperative MARL. AdaMa dynamically evaluates the contribution of actors and critic to the optimization and adaptively updates the learning rates based on their quantitative contributions. First, we examine the gain of Q-value contributed by the update of each actor. We derive the direction along which the Q-value improves the most. Thus, we can update the vector of learning rates of all actors towards the direction of maximizing the Q-value, which leads to diverse learning rates that explicitly captures the contributions of actors. Second, we consider the critic and actors are updated simultaneously. If the critic's update causes a large change of Q-value, we should give a high learning rate to the critic since it is leading the learning. However, the optimization of actors, which relies on the critic, would strug-

gle with the fast-moving target. Thus, the learning rates of actors should be reduced accordingly. On the other hand, if the critic has reached a plateau, increasing the learning rates of actors could quickly improve the actors, which further generates new experiences to boost the critic's learning. These two processes alternate during training, promoting the overall learning. Further, by incorporating the second-order approximation, we additionally capture the pairwise interaction between actors' updates so as to more accurately update the learning rates of actors towards maximizing the improvement of Q-value.

We evaluate AdaMa in four typical multi-agent cooperation scenarios, *i.e.*, *going together*, *cooperative navigation*, *predator-prey*, and *clustering*. Empirical results demonstrate that dynamically regulating the learning rates of actors and critic according to the contributions to the change of Q-value could accelerate the learning and improve the performance, which can be further enhanced by additionally considering the effect of pairwise actors' updates. The visualizations of learning rates during training clearly explain how and why AdaMa works.

## 2 RELATED WORK

**MARL.** We consider the formulation of decentralized partially observable Markov decision process (Dec-POMDP). There are $N$ agents interacting with the environment. At each timestep $t$, each agent $i$ receives a local observation $o_t^i$, takes an action $a_t^i$, and gets a shared reward $r_t$. The agents aim to maximize the expected return $\mathbb{E}\sum_{t=0}^{T}\gamma^t r_t$, where $\gamma$ is a discount factor and $T$ is the episode time horizon. Many methods (Lowe et al., 2017; Foerster et al., 2018; Rashid et al., 2018; Iqbal & Sha, 2019; Son et al., 2019) have been proposed for Dec-POMDP, which adopt centralized learning and decentralized execution (CTDE). In many of these methods, a centralized critic learns a joint Q-function by minimizing the TD-error. In training, the critic is allowed to use the information of all agents. The actors, which only have access to local information, learn to maximize the Q-value learned by the critic. In execution, the critic is abandoned and the actors act in a decentralized manner.

**Adaptive Learning Rate.** Learning rate schedules aim to reduce the learning rate during training according to a pre-defined schedule, including time-based decay, step decay, and exponential decay. The schedules have to be defined in advance and depend heavily on the type of model and problem, which requires much expert knowledge. Some optimizers, such as AdaGrad (Duchi et al., 2011), AdaDelta (Zeiler, 2012), RMSprop (Tieleman & Hinton, 2012), and Adam (Kingma & Ba, 2015), provide adaptive learning rate to ease manual tuning. AdaGrad performs larger updates for more sparse parameters and smaller updates for less sparse parameters, and other methods are derived from AdaGrad. However, these methods only deal with the gradient pattern for general optimization problems, offering no specialized way to boost multi-agent learning. WoLF (Bowling & Veloso, 2002) provides variable learning rates for stochastic games, but not for cooperation.

**Meta Gradients for Hyperparameters.** Some meta-learning methods employ hyperparameter gradients to tune the hyperparameter automatically. Maclaurin et al. (2015) utilized the reverse-mode differentiation of hyperparameters to optimize step sizes, momentum schedules, weight initialization distributions, parameterized regularization schemes, and neural network architectures. Xu et al. (2018) computed the meta-gradient to update the discount factor and bootstrapping parameter in reinforcement learning. OL-AUX (Lin et al., 2019) uses the meta-gradient to automate the weights of auxiliary tasks. The proposed AdaMa can also be viewed as a meta-gradient method for adaptive learning rates in MARL.

## 3 METHOD

In this section, we first introduce the single-critic version of MADDPG (Lowe et al., 2017), on which we instantiate AdaMa. However, AdaMa can also be instantiated on other MARL methods, and the instantiation on MAAC (Iqbal & Sha, 2019) for discrete action space is also given in Appendix A.1. Then, we use the Taylor approximation to evaluate the contributions of the critic and actors' updates to the change of Q-value. Based on the derived quantitative contributions, we dynamically adjust the direction of the vector of actors' learning rates and balance the learning rates between the critic and actors. Further, we incorporate higher-order approximation to estimate the contributions more accurately.

### 3.1 SINGLE-CRITIC MADDPG

In mixed cooperation and competition, each MADDPG agent learns an actor $\pi_i$ and a critic for the local reward. However, since the agents share the reward in Dec-POMDP, we only maintain a single shared critic, which takes the observation vector $\vec{o}$ and the action vector $\vec{a}$ and outputs the Q-value, as illustrated in Figure 1. The critic parameterized by $\phi$ is trained by minimizing the TD-error $\delta$

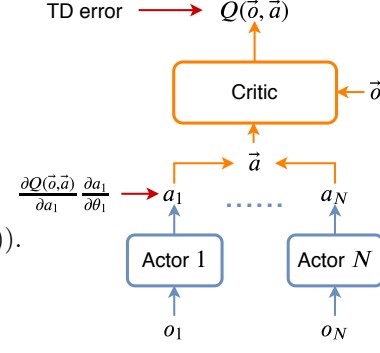

$$\mathbb{E}_{(\vec{o},\vec{a},r,\vec{o}')\sim\mathcal{D}}\left[(Q(\vec{o},\vec{a})-y)^2\right], \text{ where } y = r+\gamma Q^-(\vec{o'},\pi_i^-(o_i')).$$

$Q^-$ is the target critic, $\pi_i^-$ is the target actor, and $\mathcal{D}$ is replay buffer. Each actor $\pi_i$ (parameterized by $\theta_i$ ) is updated to maximize the learned Q-value by gradient ascent. The gradient of $\theta_i$ is

$$\frac{\partial Q(\vec{o},\vec{a})}{\partial a_i}\frac{\partial a_i}{\partial \theta_i}.$$

Figure 1: Single-Critic MADDPG

We denote the learning rates of each actor $i$ and the critic as $l_{a_i}$ and $l_c$ respectively.

### 3.2 ADAPTIVE $\vec{l_a}$ DIRECTION

First, suppose that the critic is trained and frozen, and we only update the actors. By expanding the Q-function, we can estimate the gain of Q-value contributed by actors' updates by the Taylor approximation:

$$\Delta Q = Q(\vec{o}, \vec{a} + \Delta \vec{a}) - Q(\vec{o}, \vec{a})$$

$$\approx Q(\vec{o},\vec{a}) + \sum_{i=1}^{N} \Delta a_i \frac{\partial Q(\vec{o},\vec{a})}{\partial a_i}^T - Q(\vec{o},\vec{a})$$

$$= \sum_{i=1}^{N}[\pi_i(\theta_i + l_{a_i}\frac{\partial Q(\vec{o},\vec{a})}{\partial \theta_i}) - \pi_i(\theta_i)]\frac{\partial Q(\vec{o},\vec{a})}{\partial a_i}^T$$

$$\approx \sum_{i=1}^{N} l_{a_i}\frac{\partial Q(\vec{o},\vec{a})}{\partial \theta_i}\frac{\partial a_i}{\partial \theta_i}^T\frac{\partial Q(\vec{o},\vec{a})}{\partial a_i}^T$$

$$= \sum_{i=1}^{N} l_{a_i}\frac{\partial Q(\vec{o},\vec{a})}{\partial \theta_i}\frac{\partial Q(\vec{o},\vec{a})}{\partial \theta_i}^T$$

$$= \vec{l_a} \cdot \frac{\partial Q}{\partial \theta}\frac{\vec{\partial Q}}{\partial \theta}^T.$$

Assuming the magnitude of the learning rate vector $\|\vec{l_a}\|$ is a fixed small constant $\widehat{\|\vec{l_a}\|}$, the largest $\Delta Q$ is obtained when the direction of $\vec{l_a}$ is consistent with the direction of vector $\frac{\partial Q}{\partial \theta}\frac{\vec{\partial Q}}{\partial \theta}^T$. Thus, we can softly update $\vec{l_a}$ to the direction of $\frac{\partial Q}{\partial \theta}\frac{\vec{\partial Q}}{\partial \theta}^T$ to improve the Q-value:

$$\vec{l_a} = \alpha\vec{l_a} + (1-\alpha)\widehat{\|\vec{l_a}\|}\frac{\partial Q}{\partial \theta}\frac{\vec{\partial Q}}{\partial \theta}^T / \|\frac{\partial Q}{\partial \theta}\frac{\vec{\partial Q}}{\partial \theta}^T\|$$

$$\vec{l_a} = \vec{l_a}\frac{\widehat{\|\vec{l_a}\|}}{\|\vec{l_a}\|}, \tag{1}$$

where the second line normalizes the magnitude of $\vec{l_a}$ to $\widehat{\|\vec{l_a}\|}$, and $\alpha$ is a parameter that controls the soft update. From another perspective, the update rule (1) can be seen as updating $\vec{l_a}$ by gradient ascent to increase the Q-value the most, since $\frac{\partial \Delta Q}{\partial \vec{l_a}} = \frac{\partial Q}{\partial \theta}\frac{\vec{\partial Q}}{\partial \theta}^T$.

### 3.3 ADAPTIVE $l_c$ AND $\|\vec{l_a}\|$

In the previous section, we assume that the critic is frozen. However, in MADDPG and other MARL methods, the critic and actors are trained simultaneously. Therefore, we investigate the change of Q-value by additionally considering the critic's update:

$$
\begin{aligned}
\Delta Q &= Q(\phi + \Delta\phi, \vec{o}, \vec{a} + \Delta\vec{a}) - Q(\phi, \vec{o}, \vec{a}) \\
&\approx Q(\phi, \vec{o}, \vec{a}) + \sum_{i=1}^{N} \Delta a_i \frac{\partial Q(\phi, \vec{o}, \vec{a})}{\partial a_i}^T + \Delta\phi \frac{\partial Q(\phi, \vec{o}, \vec{a})}{\partial \phi}^T - Q(\phi, \vec{o}, \vec{a}) \\
&\approx \vec{l_a} \cdot \frac{\partial Q}{\partial \theta} \frac{\partial Q}{\partial \theta}^T - l_c \frac{\partial \delta}{\partial \phi} \frac{\partial Q}{\partial \phi}^T .
\end{aligned}
$$

We can see that $\Delta Q$ is contributed by the updates of both the critic and actors. In principle, the critic's learning is prioritized since the actor's learning is determined by the improved critic. When the critic's update causes a large change of the Q-value, the critic is leading the learning, and we should assign it a high learning rate. However, the optimization of actors, which relies on the current critic, would struggle with the fast-moving target. Therefore, the actors' learning rates should be reduced. On the other hand, when the critic has reached a plateau, increasing the actors' learning rates could quickly optimize the actors, which further injects new experiences into the replay buffer to boost the critic's learning, thus promoting the overall learning. The contributions of actors' updates are always nonnegative, but the critic's update might either increase or decrease the Q-value. Therefore we use the absolute value $|\frac{\partial \delta}{\partial \phi} \frac{\partial Q}{\partial \phi}^T|$ to evaluate the contribution of critic to the change of Q-value. Based on the principles above, we adaptively adjust $l_c$ and $\|\vec{l_a}\|$ by the update rules:

$$
l_c = \alpha l_c + (1 - \alpha) l \cdot \text{clip}(|\frac{\partial \delta}{\partial \phi} \frac{\partial Q}{\partial \phi}^T| / m, \epsilon, 1 - \epsilon) \tag{2}
$$

$$
\widehat{\|\vec{l_a}\|} = l - l_c.
$$

The hyperparameters $\alpha$, $m$, $l$, and $\epsilon$ have intuitive interpretations and are easy to tune. $\alpha$ controls the soft update and $m$ controls the target value of $l_c$. The clip function and the small constant $\epsilon$ prevent the learning rate being too large or too small. Therefore, AdaMa works as follows: first update $l_c$ and get $\widehat{\|\vec{l_a}\|}$ using (2), then regulate the direction and magnitude of $\vec{l_a}$ according to (1).

As Liessner et al. (2019) pointed out, the actor should have a lower learning rate than the critic, and a high learning rate of actor leads to a performance breakdown. Also, empirically, in DDPG (Lillicrap et al., 2016) the critic's learning rate is set to 10 times higher than the actor's learning rate. However, we believe such a setting only partially addresses the problem. During training, if the learning rates of actor are always low, actors learn slowly and thus the learning is limited. Therefore, AdaMa decreases $l_c$ and increases $\|\vec{l_a}\|$ when the learning of critic reaches a plateau, which could avoid the fast-moving target and speed up the learning.

### 3.4 SECOND-ORDER APPROXIMATION

Under the first-order Taylor approximation, the actor $i$'s contribution to $\Delta Q$ is only related to the change of $a_i$, without capturing the joint effect with other agents' updates. However, when there are strong correlations between agents, the increase of the Q-value cannot be sufficiently estimated as the sum of individual contributions of each actor' update, which instead is a result of the joint update. To estimate the actors' contributions more precisely, we extend AdaMa to the second-order Taylor approximation to take pairwise agents' updates into account:

$$
\begin{aligned}
\Delta Q &= Q(\vec{o}, \vec{a} + \Delta\vec{a}) - Q(\vec{o}, \vec{a}) \\
&\approx Q(\vec{o}, \vec{a}) + \sum_{i=1}^{N} \Delta a_i \frac{\partial Q(\vec{o}, \vec{a})}{\partial a_i}^T + \frac{1}{2} \sum_{i,j=1}^{N} \Delta a_i \frac{\partial^2 Q(\vec{o}, \vec{a})}{\partial a_i \partial a_j} \Delta a_j^T - Q(\vec{o}, \vec{a}) \\
&\approx \sum_{i=1}^{N} l_{a_i} \frac{\partial Q(\vec{o}, \vec{a})}{\partial \theta_i} \frac{\partial Q(\vec{o}, \vec{a})}{\partial \theta_i}^T + \frac{1}{2} \sum_{i,j=1}^{N} l_{a_i} l_{a_j} \frac{\partial Q(\vec{o}, \vec{a})}{\partial \theta_i} \frac{\partial a_i}{\partial \theta_i}^T \frac{\partial^2 Q(\vec{o}, \vec{a})}{\partial a_i \partial a_j} \frac{\partial a_j}{\partial \theta_j} \frac{\partial Q(\vec{o}, \vec{a})}{\partial \theta_j}^T .
\end{aligned}
$$

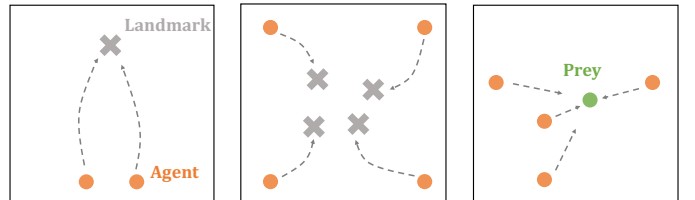

Figure 2: Illustration of experimental scenarios: going together, cooperative navigation, predator-prey, and clustering (from left to right).

As the actors are updated by the first-order gradient, we still estimate $\Delta \vec{a}$ utilizing the first-order approximation and compute the second-order $\Delta Q$ on the first-order $\Delta \vec{a}$. Then, the gradient of $l_{a_i}$ is $\frac{\partial \Delta Q}{\partial l_{a_i}} = \frac{\partial Q}{\partial \theta_i}\frac{\partial Q}{\partial \theta_i}^T + \frac{1}{2}\sum_{j=1}^{N} l_{a_j}\frac{\partial Q}{\partial \theta_i}\frac{\partial a_i}{\partial \theta_i}^T\frac{\partial^2 Q}{\partial a_i \partial a_j}\frac{\partial a_j}{\partial \theta_j}\frac{\partial Q}{\partial \theta_j}^T + \frac{1}{2}\sum_{j=1}^{N} l_{a_j}\frac{\partial Q}{\partial \theta_j}\frac{\partial a_j}{\partial \theta_j}^T\frac{\partial^2 Q}{\partial a_j \partial a_i}\frac{\partial a_i}{\partial \theta_i}\frac{\partial Q}{\partial \theta_i}^T$. Similarly, $\vec{l_a}$ can be updated as:

$$\vec{l_a} = \alpha\vec{l_a} + (1-\alpha)\widehat{\|\vec{l_a}\|}\frac{\partial \Delta Q}{\partial \vec{l_a}}/\|\frac{\partial \Delta Q}{\partial \vec{l_a}}\|, \quad \vec{l_a} = \vec{l_a}\frac{\widehat{\|\vec{l_a}\|}}{\|\vec{l_a}\|}. \tag{3}$$

## 4 EXPERIMENTS

We validate AdaMa in four cooperation scenarios with continuous observation space and continuous action space, which are illustrated in Figure 2. In these scenarios, agents observe the relative positions of other agents, landmarks, and other items, and take two-dimensional actions $\in [-1, 1]$ as physical velocity.

- *Going Together.* In the scenario, there are 2 agents and 1 landmark. The reward is $-0.5(d_i + d_j) - d_{ij}$, where $d_i$ is the distance from agent $i$ to the landmark, and $d_{ij}$ is the distance between the two agents. The agents have to go to the landmark together, avoiding moving away from each other.
- *Cooperative Navigation.* In the scenario, there are 4 agents and 4 corresponding landmarks. The reward is $-\max_i(d_i)$, where $d_i$ is the distance from agent $i$ to the landmark $i$. The slowest agent determines the reward in this scenario.
- *Predator-Prey.* In the scenario, 4 slower agents learn to chase a faster rule-based prey. Each time one of the agents collide with the prey, the agents get a reward $+1$.
- *Clustering.* In the scenario, 8 agents learn to cluster together. The reward is $-\sum d_i$, where $d_i$ is the distance from agent $i$ to the center of agents' positions. Since the center is changing along with the agents' movements, there are strong interactions between agents.

To investigate the effectiveness of AdaMa and for ablation, we evaluate the following methods:

- AdaMa adjusts $l_c$ and $\|\vec{l_a}\|$ using (2), and $\vec{l_a}$ according to (1).
- Fixed lr uses grid search to find the optimal combination of $l_c$ and $\|\vec{l_a}\|$ from $0.01$ to $0.001$ with step $0.001$. The learning rate of each agent is set to $\|\vec{l_a}\|/\sqrt{N}$.
- Adaptive $\vec{l_a}$ direction sets $l_c$ and $\|\vec{l_a}\|$ as that in Fixed lr and only adjusts the direction of $\vec{l_a}$ using (1). Additionally, Adaptive $\vec{l_a}$ direction (2nd) uses the update rule (3) for the second-order approximation.
- Adaptive $l_c$ and $\|\vec{l_a}\|$ adjusts $l_c$ and $\|\vec{l_a}\|$ using (2) and sets $l_{a_i} = \|\vec{l_a}\|/\sqrt{N}$.
- AdaGrad is an adaptive learning rate optimizer that performs larger updates for more sparse parameters and smaller updates for less sparse parameter. The initial learning rates are sets as that in Fixed lr.

Except AdaGrad, all other methods use SGD optimizer without momentum. More details about experimental settings and hyperparameters are available in Appendix A.3. We trained all the models for five runs with different random seeds. All the learning curves are plotted using mean and standard deviation.

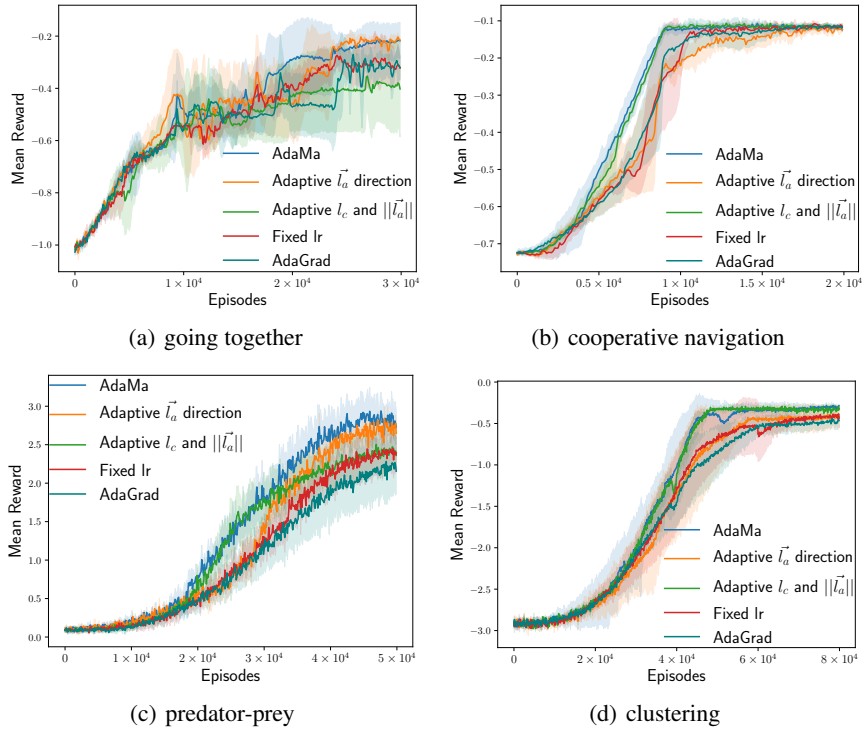

Figure 3: Learning curves in the four scenarios.

## 4.1 PERFORMANCE OF ADAPTIVE $\vec{l_a}$ DIRECTION

As shown in Figure 3(a) and 3(c), Adaptive $\vec{l_a}$ direction converges to a higher reward than Fixed lr that treats each agent as equally important. To make an explicit explanation, we visualize the normalized actors' learning rates $\vec{l_a}/\|\vec{l_a}\|$ in Figure 4 for one run and more results are available in Appendix A.4. In going together and predator-prey, the actors' learning rates fluctuate dynamically and alternately as depicted in Figure 4(a) and 4(b). An actor has a much higher learning rate than other actors in different periods, meaning that the actor is critical to the learning. The direction of $\vec{l_a}$ is adaptive to the changing contributions during the learning, assigning higher learning rates to the actors that make more contributions to $\Delta Q$. In clustering, the center is determined by all agents' positions, and the actors' updates make similar contributions to $\Delta Q$, leading to similar learning rates for the actors. That is the reason Adaptive $\vec{l_a}$ direction is not beneficial in this scenario. Moreover,

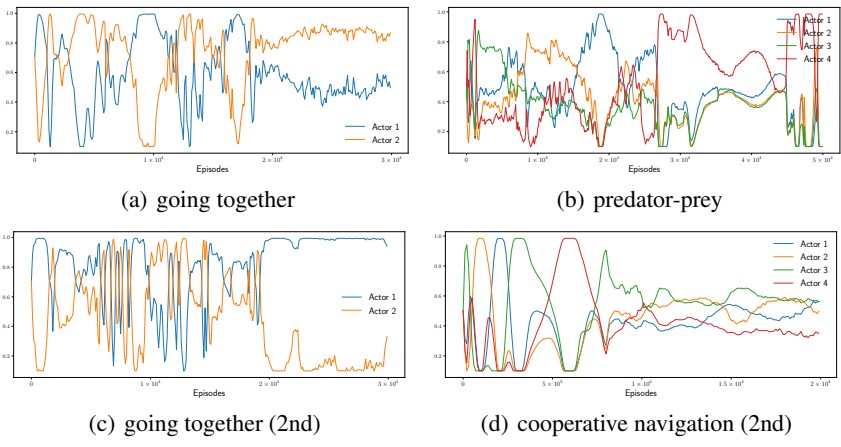

Figure 4: Normalized actors' learning rates.

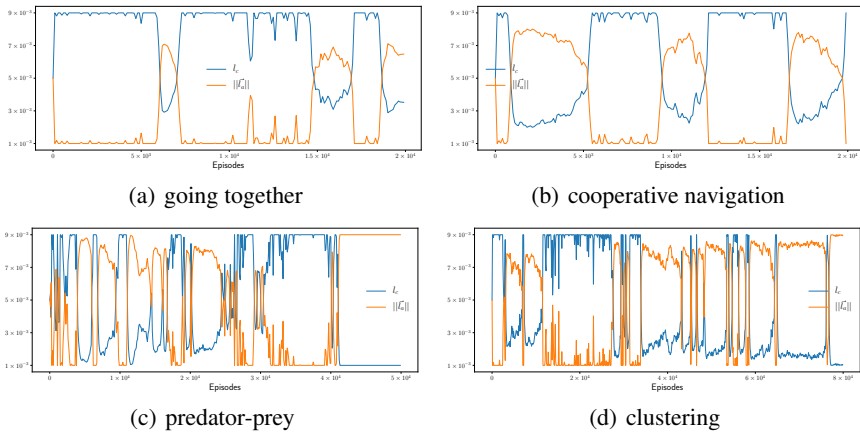

Figure 5: $l_c$ and $\|\vec{l_a}\|$.

in single-critic MADDPG, the gradient of an actor depends on the current policies of other actors. If other actors are updating at a similar rate, the update of this actor will become unstable, since the changes of others' policies are invisible and unpredictable. In our method, the agents critical to increasing the Q-value learn fast while other agents have low learning rates, which partly attenuates the instability.

## 4.2 Performance of Adaptive $l_c$ and $\|\vec{l_a}\|$

As illustrated in Figure 3(b), 3(c), and 3(d), Adaptive $l_c$ and $\|\vec{l_a}\|$ learns faster than Fixed lr. To interpret the results, we plot $l_c$ and $\|\vec{l_a}\|$ during the training in Figure 5 and find that $l_c$ and $\|\vec{l_a}\|$ rise and fall alternately and periodically. When the update of the critic impacts greatly on $\Delta Q$, *e.g.*, at the beginning with large TD-error, the fast-moving Q-value, which is the optimization target of actors, might cause a performance breakdown if the actors are also learning fast. In this situation, our method could adaptively speed up the learning of the critic and slow down the learning of actors for stability. After a while, the TD-error becomes small and makes the critic reach a plateau. According to the update rules (2), the learning of actors is accelerated whilst the learning rate of the critic falls, which keeps the target of actors stable and thus avoids the breakdown. The fast-improving actors generate new experiences, which change the distribution in the replay buffer and increase the TD-error. As a consequence, the learning rate of the critic rises again. Therefore, the learning rates of the critic and actors fluctuate alternately, promoting the overall learning continuously. In going together, the alternate fluctuation is not obvious, so Adaptive $l_c$ and $\|\vec{l_a}\|$ performs worse than Fixed lr with grid search.

Combined with the two adaptive mechanisms, AdaMa learns faster and converges to a higher reward than all other baselines in Figure 3(c). In other scenarios, AdaMa produces similar results to the

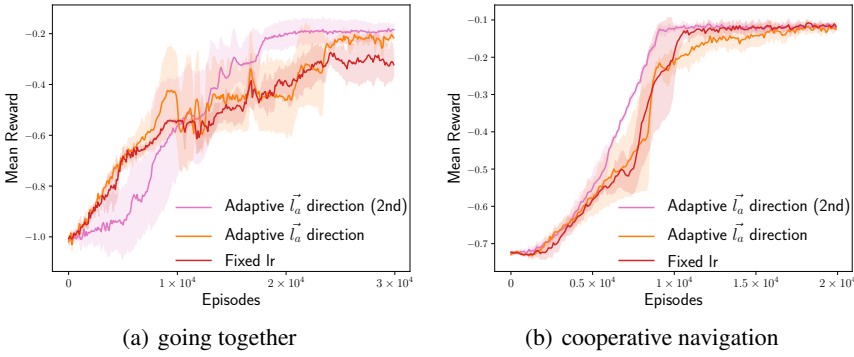

Figure 6: Learning curves with the second-order approximation.

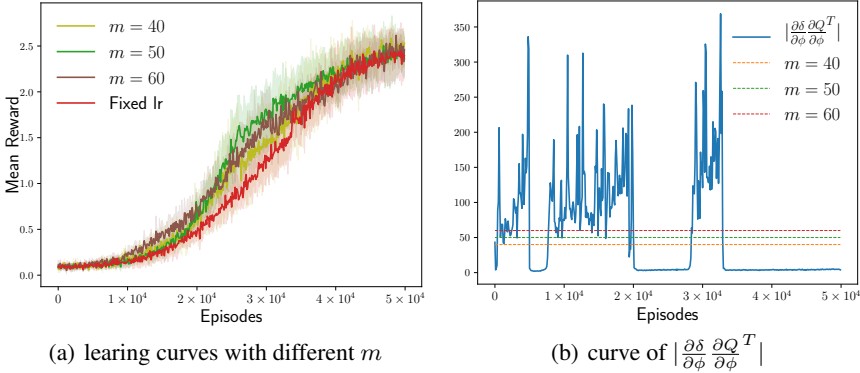

(a) learing curves with different $m$        (b) curve of $|\frac{\partial \delta}{\partial \phi} \frac{\partial Q}{\partial \phi}^T|$

Figure 7: Learning curves and visualizations with different $m$ in Predator-Prey.

mechanism that brings the main improvement. Since Fixed lr has to search 100 combinations, the cost is prohibitive. Despite adaptively adjusting the learning rates, AdaGrad does not show competitive performance, since it only focuses on the gradient pattern, ignoring the characteristics of MARL.

### 4.3 PERFORMANCE OF SECOND-ORDER APPROXIMATION

In Figure 3, the performance gain of Adaptive $\vec{l_a}$ direction is limited, which we think is attributed to that the first-order approximation is relatively rough when an actor's update affects other actors' updates. We apply the second-order approximation to Adaptive $\vec{l_a}$ direction (2nd) and find that it achieves better results as shown in Figure 6. Comparing the later episodes in Figure 4(a) and 4(c), there is a larger gap between the two actors' learning rates under the second-order approximation. This accurately reflects the later training is dominated by one actor, which is the reason for the higher reward. In Figure 4(d), there are obvious ups and downs in the learning rates of actors before convergence ($1 \times 10^4$ episodes), and after that the fluctuation becomes gentle. The second-order approximation that captures the pairwise effect of agents' updates on $\Delta Q$ obtains a more accurate update on the learning rates, which eventually leads to better performance.

### 4.4 TUNING HYPERPARAMETER $m$

The hyperparameter $m$ controls the target value of the critic's learning rate. If $m$ is too large or too small, the learning rate will reach the boundary value $\epsilon l$ or $(1 - \epsilon)l$, which destroys the adaptability and hampers the learning process. An empirical approach for tuning is setting $m$ to be the mean $|\frac{\partial \delta}{\partial \phi} \frac{\partial Q}{\partial \phi}^T|$ of the first $K$ updates in a trial run. In predator-prey, we test $m = 40, 50, 60$, among which 50 is the rounding mean value of 100 updates, and plot the results in Figure 7(a). The three settings show similar performance, revealing our method is robust to the hyperparameter $m$. Having noticed that $l_c$ and $\|\vec{l_a}\|$ change violently in Figure 5, we visualize $|\frac{\partial \delta}{\partial \phi} \frac{\partial Q}{\partial \phi}^T|$ during the training in Figure 7(b) to interpret the robustness. Since most of the time $|\frac{\partial \delta}{\partial \phi} \frac{\partial Q}{\partial \phi}^T|$ is higher than 60 or lower than 40, similar learning rate patterns is observed when $m$ is between 40 and 60, which verifies that there is high fault tolerance in $m$. Although Adaptive $l_c$ and $\|\vec{l_a}\|$ converges to a similar reward with Fixed lr, the former learns faster and is much easier to tune.

## 5 CONCLUSION

In this paper, we proposed AdaMa for adaptive learning rates in MARL. AdaMa adaptively updates the vector of learning rates of actors to the direction of maximally improving the Q-value. It also dynamically balances the learning rates between the critic and actors during learning. Moreover, AdaMa can incorporates the higher-order approximation to more accurately update the learning rates of actors. Empirically, we show that AdaMa could accelerate the learning and improve the performance in a variety of multi-agent scenarios.

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

# A APPENDIX

## A.1 INSTANTIATING ADAMA ON MAAC

AdaMa could be applied to other multi-agent actor-critic methods, modified according to the gradient computation. We present how to instantiate AdaMa on MAAC (Iqbal & Sha, 2019). In discrete action space, $\Delta\vec{a}$ does not meet the assumption of the Taylor approximation. But in practice, it is feasible to learn the value function $Q(\phi, \vec{o}, \vec{\pi})$, taking as input the action distribution $\pi_i$ instead of the action $a_i$. Since the update of the actor is different from that in MADDPG, the change of Q-value is written as

$$
\begin{aligned}
\Delta Q &= Q(\phi + \Delta\phi, \vec{o}, \vec{\pi} + \Delta\vec{\pi}) - Q(\phi, \vec{o}, \vec{\pi}) \\
&\approx Q(\phi, \vec{o}, \vec{\pi}) + \sum_{i=1}^{N} \Delta\pi_i \frac{\partial Q(\phi, \vec{o}, \vec{\pi})}{\partial \pi_i}^T + \Delta\phi \frac{\partial Q(\phi, \vec{o}, \vec{\pi})}{\partial \phi}^T - Q(\phi, \vec{o}, \vec{\pi}) \\
&= \sum_{i=1}^{N} [\pi_i(\theta_i + l_{a_i} \frac{\partial \log \pi_i A_i}{\partial \theta_i}) - \pi_i(\theta_i)] \frac{\partial Q(\phi, \vec{o}, \vec{\pi})}{\partial \pi_i}^T + \Delta\phi \frac{\partial Q(\phi, \vec{o}, \vec{\pi})}{\partial \phi}^T \\
&\approx \sum_{i=1}^{N} l_{a_i} \frac{\partial \log \pi_i A_i}{\partial \theta_i} \frac{\partial \pi_i}{\partial \theta_i}^T \frac{\partial Q(\phi, \vec{o}, \vec{\pi})}{\partial \pi_i}^T - l_c \frac{\partial \delta}{\partial \phi} \frac{\partial Q(\phi, \vec{o}, \vec{\pi})}{\partial \phi}^T \\
&= \sum_{i=1}^{N} l_{a_i} \frac{\partial \log \pi_i A_i}{\partial \theta_i} \frac{\partial Q}{\partial \theta_i}^T - l_c \frac{\partial \delta}{\partial \phi} \frac{\partial Q}{\partial \phi}^T .
\end{aligned}
$$

$A_i$ is the advantage function of agent $i$' proposed in MAAC. Rewritting the gradient of $l_{a_i}$ as $\frac{\partial \Delta Q}{\partial l_{a_i}} = \frac{\partial \log \pi_i A_i}{\partial \theta_i} \frac{\partial Q}{\partial \theta_i}^T$, the AdaMa implementation on MAAC is the same as that on MADDPG.

## A.2 ADAMA ALGORITHM

For completeness, we provide the AdaMa algorithm on MADDPG below.

---
**Algorithm 1** AdaMa on MADDPG
---
1: Initialize critic network $\phi$, actor networks $\theta_i$, and target networks
   Initialize the learning rates $l_c$ and $\vec{l_a}$
   Initialize replay buffer $\mathcal{D}$
2: **for** episode = $1, \ldots, \mathcal{M}$ **do**
3:    **for** $t = 1, \ldots, \mathcal{T}$ **do**
4:       Select action $a_t^i = \pi_i(o_t^i) + \mathcal{N}_t^i$ for each agent $i$
5:       Execute action $a_t^i$, obtain reward $r_t$, and get new observation $o_{t+1}^i$ for each agent $i$
6:       Store transition $(\vec{o}_t, \vec{a}_t, r_t, \vec{o}_{t+1})$ in $\mathcal{D}$
7:    **end for**
8:    Sample a random minibatch of transitions from $\mathcal{D}$
      Adjust $l_c$ and $\|\vec{l_a}\|$ by $l_c = \alpha l_c + (1-\alpha) l \cdot \text{clip}(|\frac{\partial \delta}{\partial \phi} \frac{\partial Q}{\partial \phi}^T|/m, \epsilon, 1 - \epsilon), \widehat{\|\vec{l_a}\|} = l - l_c$
      Adjust $\vec{l_a}$ by $\vec{l_a} = \alpha \vec{l_a} + (1-\alpha) \widehat{\|\vec{l_a}\|} \frac{\partial \Delta Q}{\partial \vec{l_a}} / \|\frac{\partial \Delta Q}{\partial \vec{l_a}}\|, \vec{l_a} = \vec{l_a} \frac{\widehat{\|\vec{l_a}\|}}{\|\vec{l_a}\|}$
      Update the critic $\phi$ by $\phi = \phi - l_c \frac{\partial \delta}{\partial \phi}$, where $\delta$ is the TD-error
      Update the actor $\theta_i$ by $\theta_i = \theta_i + l_{a_i} \frac{\partial Q(\vec{o}, \vec{a})}{\partial a_i} \frac{\partial a_i}{\partial \theta_i}$ for each agent
      Update the target networks
9: **end for**
---

## A.3 EXPERIMENTAL SETTINGS AND HYPERPARAMETERS

In each task, the experimental settings and hyperparameters are summarized in Table 1. Initially, we set $l_c = \|\vec{l_a}\|$ and $l_{a_i} = \|\vec{l_a}\|/\sqrt{N}$ in AdaMa. For exploration, we add random noise to the action

$(1-\varepsilon)a_i + \varepsilon\eta$, where the uniform distribution $\eta \in [-1, 1]$. We anneal $\varepsilon$ linearly from $1.0$ to $0.1$ over $10^4$ episodes and keep it constant for the rest of the learning. We update the model every episode and update the target networks every 20 episodes.

Table 1: Hyperparameters

| Hyperparameter | Going Together | Cooperative Navigation | Predator-Prey | Clustering |
|---|---|---|---|---|
| horizon ($T$) | 10 | 6 | 20 | 10 |
| discount ($\gamma$) | 0.96 | 0.9 | 0.97 | 0.95 |
| replay buffer size | $5 \times 10^5$ | $5 \times 10^5$ | $1 \times 10^6$ | $1 \times 10^6$ |
| $l_c$ (grid search) | $8 \times 10^{-3}$ | $9 \times 10^{-3}$ | $7 \times 10^{-3}$ | $8 \times 10^{-3}$ |
| $\|\vec{l_a}\|$ (grid search) | $3 \times 10^{-3}$ | $2 \times 10^{-3}$ | $2 \times 10^{-3}$ | $1 \times 10^{-3}$ |
| batch size | | 1024 | | |
| MLP units | | $(64, 64)$ | | |
| MLP activation | | ReLU | | |
| $m$ | 10 | 5 | 50 | 80 |
| $\alpha$ | | 0.99 | | |
| $l$ | | $1 \times 10^{-2}$ | | |
| $\epsilon$ | | 0.1 | | |

## A.4 ADDITIONAL RESULTS

Here, we show the visualizations in the other four runs. Similar curve patterns are observed in all runs of each scenario.

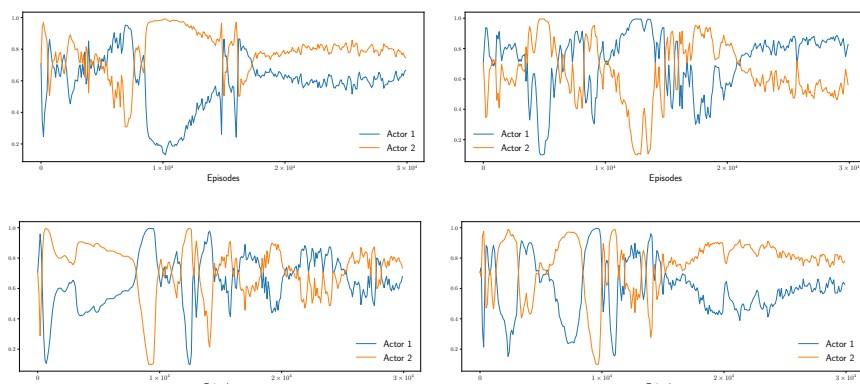

Figure 8: Normalized actors' learning rates in going together

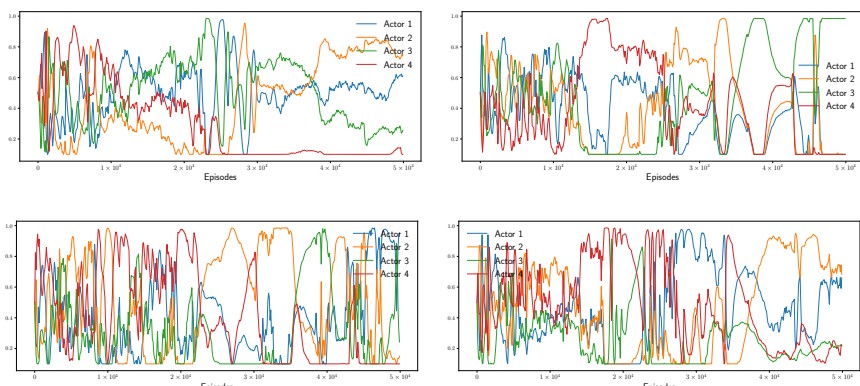

Figure 9: Normalized actors' learning rates in predator-prey

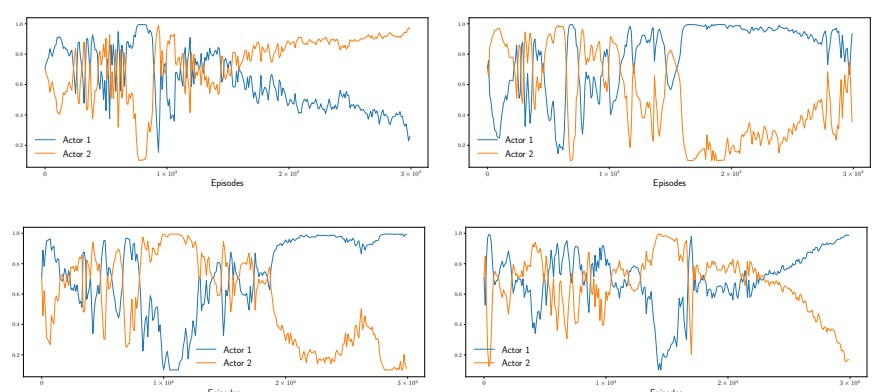

Figure 10: Normalized actors' learning rates in going together (2nd)

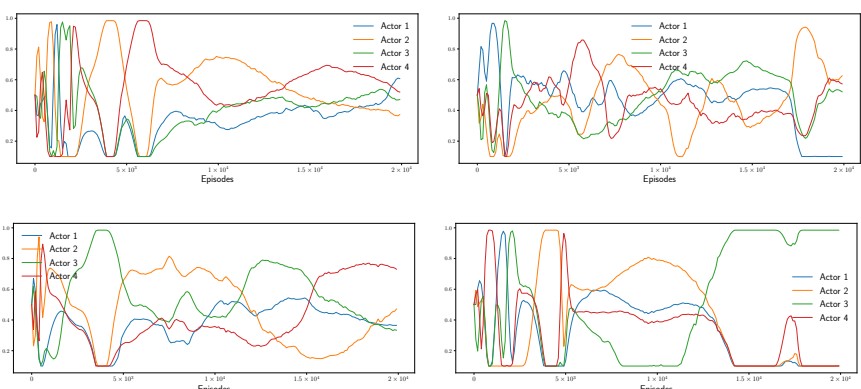

Figure 11: Normalized actors' learning rates in cooperative navigation (2nd)

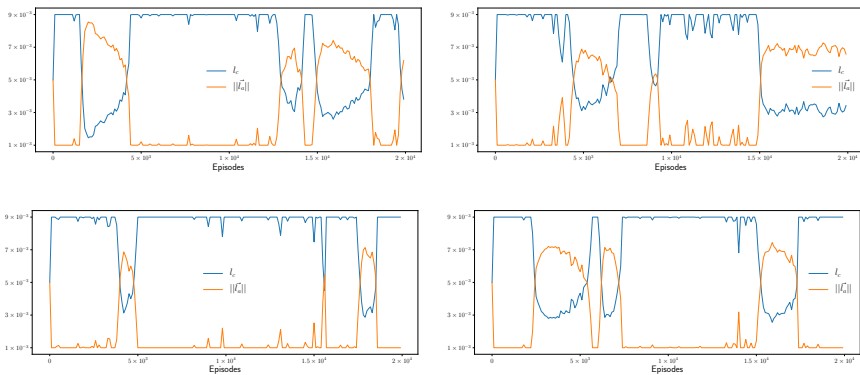

Figure 12: $l_c$ and $\|\vec{l_a}\|$ in going together.

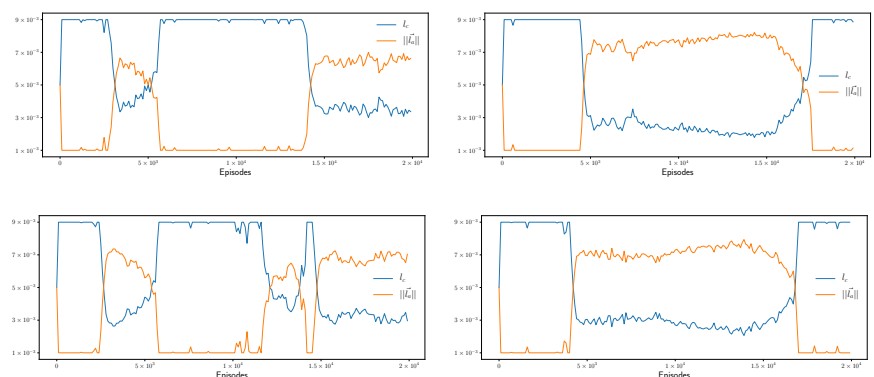

Figure 13: $l_c$ and $\|\vec{l_a}\|$ in cooperative navigation.

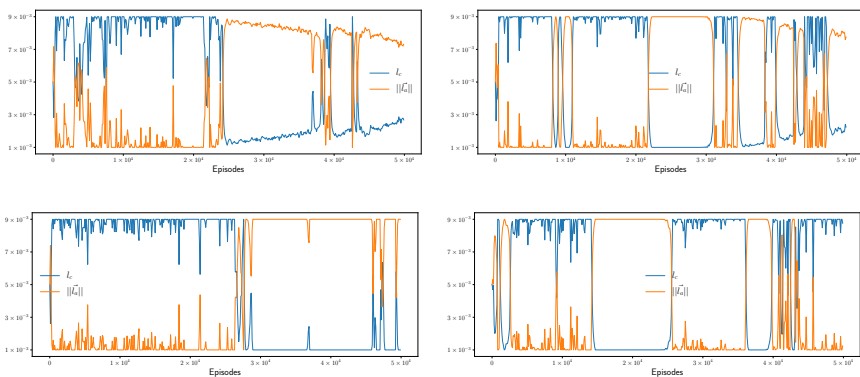

Figure 14: $l_c$ and $\|\vec{l_a}\|$ in predator-prey.

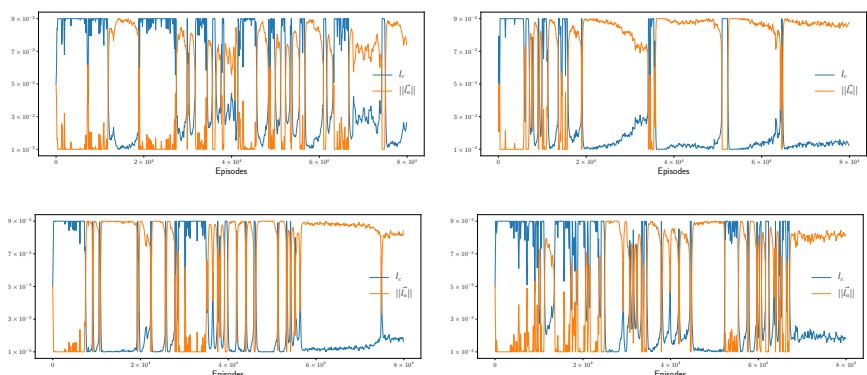

Figure 15: $l_c$ and $\|\vec{l_a}\|$ in clustering.

