# OpenReview forum: "Adaptive Learning Rates for Multi-Agent Reinforcement Learning"
_ICLR.cc/2021/Conference — Reject_

### Official Review · AnonReviewer4 · 2020-10-13
**The paper studies adaptive learning rate for Actor-Critic style MARL algorithm. I have several questions:**

**Rating:** 5
**Confidence:** 2

**Review:**

The paper studies adaptive learning rate for Actor-Critic style MARL algorithm. I have several questions:

1. How is the adaptive learning rate related to MARL? It seems this is just a general improvement for actor-critic methods?
2. This is closely related to the first question. Some optimization algorithms with adaptive learning ate are mentioned in the related work section. However, it is not clear why these methods cannot solve the problem and we need new techniques. Or, what structure in MARL makes it possible to achieve further improvement? Is this possible for, say single-agent setting?
3. I am not very familiar with the Dec-POMDP setting so I have a stupid question: shouldn't the policy a function of the state instead of the observation? Otherwise, we will need to work with the significantly larger observation space instead of the state space?

---

> ### Author Response · Authors · 2020-11-16
> **Responses to Review #4**
>
> Adaptive $\vec{l_a}$ Direction is proposed to update the learning rates of different actors to improve the joint Q-value maximally, and it is not compatible with single-agent environments since there is only *one* agent. However, Adaptive $l_c$ and $\|\vec{l_a}\|$ could be applied to other general actor-critic methods, which expands the implications of our contribution.
>
> For the existing optimizers, AdaGrad decreases the learning rate monotonically and performs larger updates for more sparse parameters. AdaDelta and RMSprop adjust AdaGrad to make the learning rate decreasing not too aggressive. They all reduce the learning rate along the training based only on the pattern of gradients, without considering the convergence of Q-value and the improvement of policies. AdaMa updates the vector of learning rates of all actors towards the direction of maximizing the $Q$ value the most. Moreover, in a single-agent setting, existing optimizers independently work on the actor and the critic, without considering the relation between them. Adaptive $l_c$ and $\|\vec{l_a}\|$ dynamically balances the learning rates between the critic and actor according to their varying effects on the overall learning in order to speed up the learning and avoid the performance breakdown.
>
> In Dec-POMDP, the agents could only obtain the partial observation of the global state and make decisions based on the partial observation. If the partial observation is too limited, we could use the history of observations to obtain more information about the state [1][2].
>
> [1] Hausknecht and Stone, Deep Recurrent Q-Learning for Partially Observable MDPs, arXiv:1507.06527, 2015.
>
> [2] Rashid et al., QMIX: Monotonic Value Function Factorisation for Deep Multi-Agent Reinforcement Learning, ICML'18.

---

### Official Review · AnonReviewer1 · 2020-10-28

**Rating:** 4
**Confidence:** 2

**Review:**

---
Summary

This paper studies automatic learning rate tuning in Multi-Agent Reinforcement Learning (MARL). It proposes AdaMa, an algorithm which balances the learning rates of actors and the critic, and can also make use of the second-order information. Experiments show that AdaMa can learn agents faster.


---
Writing Quality

The paper is not well-written as I don't fully understand it. Notations are used without definition ($\delta$ in Section 3.3) and inconsistent (some $Q$ has arrow above it and some doesn't) , and $\frac{\partial Q}{\partial \theta}$ has a confusing shape. Simpler symbols can also be used for readability (e.g., $\widehat{\|\overrightarrow{l_a}\|}$ can be replaced by something like $\eta_a$). The derivation of $\Delta Q$ can be deferred to appendix.

I'd suggest the authors to make an algorithm box for the full algorithm. Equation (1), (2) and (3) might be confusing, as they're describing algorithms, not mathematical equations.


---
Comments


I'm not fully convinced. The figures indeed show that AdaMa uses different LR for different actor. Why is this helpful? Why should we slow down the learning of an actor if it's already good?

How does Adam, one of the most widely used optimizer, work?

---

> ### Author Response · Authors · 2020-11-16
> **Responses to Review #1**
>
> We adaptively update the learning rates of actors to maximally improve the Q-value and constrain the magnitude of the learning rate vector $\|\vec{l_a}\|$ to be a fixed small constant since too large learning rates would lead to a performance breakdown in general deep learning and especially actor-critic algorithms, and that is why some optimizers decrease the learning rates along with the training. In single-critic MADDPG, the gradient of an actor depends on the current policies of other actors. The update will become unstable if other actors are updating fast since the changes of other agents are invisible and unpredictable. Therefore, we slow down the learning of good agents which make little contribution to $\Delta Q$, in order to both avoid the performance breakdown and make other agents' updates stable.
>
> We have tried other optimizers: Adam and RMSprop and found the AdaGrad achieves the best performance in the experimental scenarios, so we select AdaGrad as the baseline of general optimizers. Moreover, the comparison with Adam is unfair since Adam contains Momentum, which is not included in our method and other baselines.
>
> Please note that $\delta$ is defined in the first paragraph in page 3. The arrow indicates a vector over agents.
>
> The formal description of AdaMa is given in Appendix A.2. Please see the revision.

---

### Official Review · AnonReviewer3 · 2020-10-29
**Review for "Adaptive Learning Rates for Multi-Agent Reinforcement Learning"**

**Rating:** 4
**Confidence:** 2

**Review:**

This paper proposes an algorithm called AdaMa for multi-agent reinforcement learning (MARL). Based on the contribution of the critic and actors, the algorithm adopts adaptive learning rates. Numerical experiments are provided in four cooperation scenarios to show the performance of AdaMa.


In my view, the paper currently falls below the bar for an ICLR publication. The detailed comments are as follows:

- The key concern about this paper is the lack of rigorous analysis. The contents in Section 3 is largely based on heuristic approximations. There is no rigorous analysis, statement or proof. And there is theoretical guarantees of the performance of the algorithm with respect to the sample complexity or time complexity.

- In addition, there is no comparison with state-of-the-art algorithms, which limits the contribution. It would be much better if authors can include this either in terms of theory or numerical experiments.

- The paper is not well written. For example, there is no formal description of the proposed algorithm.

---

> ### Author Response · Authors · 2020-11-16
> **Responses to Review #3**
>
> General convergence analysis in deep MARL is extremely hard if not impossible. As also pointed by [1], "the theoretical analysis of deep MARL is an almost uncharted territory." Our work focuses on adaptive learning rates in deep MARL, and theoretical analysis will be investigated in future work.
>
> To the best of my knowledge, AdaMa is the first method for adaptive learning rates in multi-agent cooperation. We do not believe there is existing state-of-the-art algorithms in this area.
>
> The formal description of AdaMa is given in Appendix A.2. Please see the revision.
>
>
> [1] Zhang et al., Multi-agent reinforcement learning: A selective overview of theories and algorithms, arXiv:1911.10635, 2019.

---

### Official Review · AnonReviewer2 · 2020-11-02

**Rating:** 5
**Confidence:** 3

**Review:**

Summary:


The paper proposes a new algorithm for multi-agent reinforcement learning (MARL) that adaptively picks learning rates for actor and critic. Specifically, the learning rates are updated to directions maximally affecting the Q-function, and the algorithm dynamically balances the learning rates between actor and critic. In numerical studies, the authors illustrate the efficiency of their method via four toy experimental scenarios and intuitively explain the underlying mechanism.


------------------------------------------------------------------------------------------


Pros:

+ The choice of learning rates in MARL is an interesting and important issue.
+ The paper is well written. The methodology part is clearly organized and easy to follow. The learning rate balance between actor and critic is well motivated.
+ The numerical experiments are well designed. Each model represents a different cooperation mode. The results are well presented.


------------------------------------------------------------------------------------------

Cons:

- The numerical results are not satisfying. The improvement in AdaMa is incremental. In Figure 3 (a), (b) and (d), AdaMa has similar performance with Fixed lr (fixed learning rate).
- Currently each model is trained for 5 runs. More experiments are needed for more reliable results.
- The authors should evaluate the performance of AdaMa in more practical models in addition to these four toy examples.

---

> ### Author Response · Authors · 2020-11-16
> **Responses to Review #2**
>
> Although AdaMa has similar performance with Fixed lr in some scenarios, Fixed lr has to use grid search to find the optimal combination of $l_c$ and $\|\vec{l_a}\|$ from $0.01$ to $0.001$ with step $0.001$, which costs one hundred times more computation than AdaMa. AdaMa could achieve better performance with a very little tuning. From the perspective of both performance and cost, we believe the improvement is *significant*.
>
> Five runs with different random seeds are commonly used in many recent MARL papers [1][2][3][4].
>
> The experimental scenarios are modified from the popular multi-agent environment MPE [5], which is the testbed of the backbone algorithm MADDPG. There are $8$ agents in Clustering, which should not be considered as a small scale for MADDPG.
>
>
> [1] Wang et al., ROMA: Multi-Agent Reinforcement Learning with Emergent Roles, ICML'20.
>
> [2] Singh et al., Individualized Controlled Continuous Communication Model for Multiagent Cooperative and Competitive Tasks, ICLR'19.
>
> [3] Du et al., LIIR: Learning Individual Intrinsic Reward in Multi-Agent Reinforcement Learning, NeurIPS'19.
>
> [4] Das et al., TarMAC: Targeted Multi-Agent Communication, ICML'19.
>
> [5] https://github.com/openai/multiagent-particle-envs

---

### Official Review · AnonReviewer5 · 2020-11-06
**reasonable idea but the study is not thorough**

**Rating:** 5
**Confidence:** 4

**Review:**

This paper proposed AdaMa, which can automatically use adaptive learning rates for each agent in cooperative Multi-Agent Reinforcement Learning (MARL). AdaMa calculated the learning rate of each actor and critic according to their contributions of locally increasing value functions.  Simple experiments using toy examples show that the proposed AdaMa method can improve fixed learning rate method and other heuristics.

Pros:
1. The topic and idea of using adaptive learning rates to avoid hand-tuning are quite interesting and important. I think the related topics are worth investigating.
2. The proposed AdaMa method looks reasonable to me, at least from an intuitive perspective.
3. Experimental results also look promising.

Cons:
1. The experiments look simple, and the improvements seem to be incremental rather than significant (please correct me if I misunderstood or missed something). I think more experiments on larger scale tasks are needed to make the effectiveness of the proposed method convincing.
2. Using first- and second-order Taylor expansion to obtain the best possible learning rates seems to be a reasonable idea. However, I think some more rigorous theoretical understanding is worth pursuing to show the benefits of the proposed method in a more convincing way, e.g., the speed of convergence for fixed and adaptive learning rate methods.

Overall, I found the problem and the idea important and interesting. The proposed method is intuitively reasonable and verified by small scale experiments. However, the proposed method is not convincing in terms of the lack of larger-scale experiments or theoretical results.

---

> ### Author Response · Authors · 2020-11-16
> **Responses to Review #5**
>
> The experimental scenarios are modified from the popular multi-agent environment MPE [1], which is the testbed of our backbone algorithm MADDPG. There are $8$ agents in Clustering, which should not be considered as a small scale for MADDPG. The experiments look simple, but they take really a long time for agents to learn the optimal policies. Because the reward functions are strongly related to all agents, and a small change in one agent's policy would greatly influence the cumulative reward.
>
> Although the improvements of AdaMa seem to be not significant in some scenarios, the baseline Fixed lr has to use grid search to find the optimal combination of $l_c$ and $\|\vec{l_a}\|$ from $0.01$ to $0.001$ with step $0.001$, which costs one hundred times more computation than AdaMa. AdaMa could achieve better performance with a very little tuning. From the perspective of both performance and cost, we believe the improvement is *significant*.
>
> General convergence analysis in deep MARL is extremely hard if not impossible. As also pointed by [2], "the theoretical analysis of deep MARL is an almost uncharted territory." Our work focuses on adaptive learning rates in deep MARL, and theoretical analysis will be investigated in future work.
>
> [1] https://github.com/openai/multiagent-particle-envs
>
> [2] Zhang et al., Multi-agent reinforcement learning: A selective overview of theories and algorithms, arXiv:1911.10635, 2019.

---

### Author Response · Authors · 2020-11-16
**To all the reviewers**

We have posted the responses to each reviewer and hope they could address all the comments of the reviewers. From the reviews, we acknowledged that some of the reviewers might not be familiar with deep MARL. But, we are willing to discuss any comments and concerns and help the reviewers fully understand the merits of the paper.

---

### Decision · Program_Chairs · 2021-01-07
**Final Decision**

**Decision:**

Reject

**Comment:**

This paper investigates how to deploy adaptive learning rates in multi-agent RL (MARL). In particular, the learning rates are adaptively chosen based on which directions maximally affect the Q-function, and take into account the interplay and balance between the actors and the critics. The topic is certainly of great interest when designing fast-convergent MARL algorithms. However, the reviewers point out the inadequacy and insufficiency of empirical gains in the reported experiments. Also, larger-scale experimental settings are needed in order to provide more convincing evidence about the practical benefits of the proposed scheme.